# Modification of Thin Film Composite Membrane by Chitosan–Silver Particles to Improve Desalination and Anti-Biofouling Performance

**DOI:** 10.3390/membranes12090851

**Published:** 2022-08-31

**Authors:** María Magdalena Armendáriz-Ontiveros, Yedidia Villegas-Peralta, Julia Elizabeth Madueño-Moreno, Jesús Álvarez-Sánchez, German Eduardo Dévora-Isiordia, Reyna G. Sánchez-Duarte, Tomás Jesús Madera-Santana

**Affiliations:** 1Departamento de Ciencias del Agua y Medio Ambiente, Instituto Tecnológico de Sonora, 5 de Febrero 818 Sur, Ciudad Obregón 85000, Mexico; 2Centro de Investigación y Desarrollo A.C., Hermosillo 83304, Mexico

**Keywords:** chitosan, Ag, reverse osmosis membranes, bio-fouling, desalination

## Abstract

Reverse osmosis (RO) desalination is a technology that is commonly used to mitigate water scarcity problems; one of its disadvantages is the bio-fouling of the membranes used, which reduces its performance. In order to minimize this problem, this study prepared modified thin film composite (TFC) membranes by the incorporation of chitosan–silver particles (CS–Ag) of different molecular weights, and evaluated them in terms of their anti-biofouling and desalination performances. The CS–Ag were obtained using ionotropic gelation, and were characterized by Fourier transform infrared spectroscopy (FTIR), high-resolution scanning electron microscopy (HR-SEM), energy-dispersive X-ray spectroscopy (EDX), thermogravimetric analysis (TGA) and dynamic light scattering (DLS). The modified membranes were synthetized by the incorporation of the CS–Ag using the interfacial polymerization method. The membranes (MCS–Ag) were characterized by Fourier transform infrared spectroscopy (FTIR), atomic force microscopy (AFM) and contact angle. Bactericidal tests by total cell count were performed using *Bacillus halotolerans* MCC1, and anti-adhesion properties were confirmed through biofilm cake layer thickness and total organic carbon (%). The desalination performance was defined by permeate flux, hydraulic resistance, salt rejection and salt permeance by using 2000 and 5000 mg L^−1^ of NaCl. The MCS–Ag-L presented superior permeate flux and salt rejection (63.3% and 1% higher, respectively), as well as higher bactericidal properties (76% less in total cell count) and anti-adhesion capacity (biofilm thickness layer 60% and total organic carbon 75% less, compared with the unmodified membrane). The highest hydraulic resistance value was for MCS–Ag-M. In conclusion, the molecular weight of CS–Ag significantly influences the desalination and the antimicrobial performances of the membranes; as the molecular weight decreases, the membranes’ performances increase. This study shows a possible alternative for increasing membrane useful life in the desalination process.

## 1. Introduction

Water is the essential liquid for life on earth [1]; however, in recent decades, fresh water scarcity has rapidly worsened worldwide [2,3] due to anthropogenic contamination, poor water management, overexploitation and increased demand because of population growth, etc. [4]. This situation worsens in lower-income countries that cannot afford freshwater production technologies [5]. Reverse osmosis (RO) desalination is a technology that can help mitigate the water scarcity problem [6]. Since, it provides an available and reliable alternative to fresh water sources at any time of the year. However, the production costs of reverse osmosis desalination are high due to its high consumption of energy, making it an unaffordable technology for lower-income regions. Moreover, reverse osmosis membranes face the critical challenge of high fouling tendencies by different compounds such as inorganic, organic, scaling and biological (bio-fouling), which are responsible for 25–50% of total operational costs [7].

Bio-fouling involves attachment, growth, reproduction and proliferation of microorganisms (MOS) on the surfaces of the reverse osmosis membranes; this causes water production reduction, increased chemical cleaning frequency, increased operating pressure, and hence increased energy consumption [8]. Thus, there is significant practical interest to develop efficient and cost-effective approaches to avoid reverse osmosis membrane bio-fouling.

In an attempt to reduce MOS growth on reverse osmosis membranes, much research has focused on applying novel materials with biocide effects. For example, El-Gendi et al. [9] synthesized a membrane with cellulose acetate/graphene/Ag nanoparticles and/or Cu nanorod composites in order to improve membrane desalination and bio-fouling properties. The membranes exhibited antibacterial activity against *Enterococcus facium, Staphylococcus aureus, Escherichia coli, Pseudomonas aeruginosa* and *Salmonella*. Hamdy and Taher [10] added graphene oxide (GO) to a thin-film composite (TFC) membrane. They enhanced membrane antibacterial activity against Escherichia coli and Staphylococcus aureus, as well as improved membrane hydrophilicity, water flux, permeance, salt rejection and chlorine resistance. Morsy et al. [11] prepared a membrane with nanocelluloses/cellulose acetate extracted from rice straw waste. They used egg albumin as a protein feed solution to foul membranes, and they found protein resistance and adsorption abilities of membranes which are prerequisites for a surface to resist microbial adhesion. Armendariz et al. [12] coated a commercial reverse osmosis membrane surface with graphene oxide-iron nanoparticles (FeNPs) prepared by the immersion method, and found a reduction in biofilm layer thickness on the membrane surface. Rodriguez et al. [13] modified TFC membranes, incorporating multidimensional graphene oxide into the polyamide layer, and found that they yielded high desalination performance, high bactericidal properties and anti-adhesion capacity against *Escherichia coli.*

Chitosan (CS) is a natural copolymer that has begun to be used in desalination membranes, due to its reactive hydroxyl and amino groups that provide reverse osmosis membrane hydrophilic properties. For example, Shakeri et al. [14] fabricated a hydrophilic chitosan forward osmosis (FO) membrane using interfacial polymerization, and they found remarkably higher hydrophilicity, water permeation and salt rejection properties compared to a commercial thin film composite membrane. Moreover, Mehta et al. [15] modified the active layer of a commercial reverse osmosis membrane with chitosan-glutaraldehyde with the dipping method. First, they activated the polyamide (PA) layer with sodium hypochlorite, then submerged the membrane in a chitosan solution, followed by a glutaraldehyde solution. They reported a 180% increase in permeate flux with about a 2.7% increase in divalent ion rejection as compared to a virgin thin film composite reverse osmosis membrane. In addition, Hegab et al. [16] modified the PA of a commercial reverse osmosis membrane with graphene oxide and chitosan. They tested the modified membrane fouling resistance through a cross-flow system using bovine serum albumin organic as a foulant solution. They found less flux decline in modified membranes than the unmodified one.

On the other hand, chitosan is obtained by alkaline deacetylation of chitin, which is the main component of the exoskeleton of crustaceans, such as shrimps [15,17]; thus, it is an eco-friendly material since it uses organic waste for its synthesis. Its molecular weight and degree of deacetylation influence the physical and chemical properties of membranes [18]. Therefore, chitosan materials offer many opportunities to modify the physicochemical properties of membranes. For example, chitosan can help reduce fouling because it can adsorb organic matter without losing permeance [19], and has also been shown to have bactericidal or bacteriostatic properties [20]; thus, it may be useful in reducing bio-fouling in reverse osmosis membranes. At the time of writing, few published studies assessed bio-fouling on reverse osmosis membranes using chitosan. For example, Raza et al. [21] synthesized a polymeric chitosan membrane using polyethylene glycol and tetraethylorthosilicate as a cross-linker. They found high antibacterial activity against *Escherichia coli*; however, the flux and salt rejection results were low in comparison to a polyamide reverse osmosis membrane. Moreover, El-Ghaffar et al. [22] prepared reverse osmosis membranes of cellulose acetate with chitosan nanoparticles using the phase-inversion technique. Membranes were evaluated in a dead-end filtration system, which showed a 200% and 5.6% increase in permeate flux and salt rejection, respectively. The membrane containing chitosan showed enhancement in resistance to bacterial attack (*Escherichia coli*). Furthermore, Kayani et al. [23] synthetized chitosan and polyethylene glycol (PEG-600) membranes, and found higher water permeance as well as salt rejection flux and salt rejection (80% and 40.4%, respectively), in addition to antibacterial properties against *Escherichia coli.*

Moreover, chitosan’s antimicrobial properties have been extensively studied in combination with some nanocomposites (NPs). Nevertheless, these studies with chitosan and nanocomposites have been performed in health, environmental protection applications and ultrafiltration membranes [17,24,25,26]. For example, Kumar-Krishnan et al. [27] tested the antibacterial activity of chitosan–silver nanocomposites and chitosan–silver nanoparticles against Gram-positive *Staphylococcus aureus* and Gram-negative *Escherichia coli*. They found a higher antibacterial potency with chitosan–silver nanoparticles. Govindan et al. [28] synthesized and characterized a chitosan–silver nanocomposite, which exhibited good antimicrobial and antitumor properties. In addition, Tripathi et al. [29] prepared a film with a chitosan–silver oxide encapsulated nanocomposite, and tested its antibacterial activity against *Escherichia coli*, *Staphylococcus aureus*, *Bacillus subtilis* and *Pseudomonas aeruginosa*. Further, most bio-fouling tests reported in the literature for assessing modified thin film composite polyamide membranes with chitosan either used *Escherichia coli* or *Pseudomonas aeruginosa*, which are not bacteria that typically cause bio-fouling problems in reverse osmosis membranes. Moreover, many membranes have been synthesized with different types of antimicrobial materials; however, the permeate flux results are lower than those for membranes available in the market [12,30], which may make this approach impractical for the desalination industry.

On the other hand, the modification of thin film composite polyamide reverse osmosis membranes (TFC RO) by incorporating silver nanoparticles (AgNPs) has been extensively studied and different strategies for their incorporation have been proposed, since it is a highly effective compound for reducing bacterial growth on the membrane surface. Although it is an expensive compound, a small amount of silver on reverse osmosis membranes can help the desalination process. Furthermore, its properties can be intensified if it is combined with a hydrophilic material, such as chitosan. Hence, chitosan–silver particles (CS–Ag) appear to be an ideal material for sustainable and unharmful by-products to the long-existing TFC PA membranes. For these reasons, chitosan–silver compound is an attractive alternative for mitigating the problem of bio-fouling on reverse osmosis membranes [31]. Nonetheless, at the time of writing there were no reported studies of the application of chitosan–silver particles on reverse osmosis membranes for desalination using interfacial polymerization methodology.

Chitosan may be one of the most promising materials for desalination operations, given their simple synthesis process, low secondary pollution risk and relative availability at low cost compared to other nanomaterials used in reverse osmosis membranes. Given that such a coating would be an attractive alternative for the desalination industry, the research presented in this paper aimed to synthesize a reverse osmosis membrane with chitosan–silver of different molecular weights, in order to study their effect and potential to enhance desalination performance and reduce bio-fouling problems in desalination plants. The present study constitutes the first antibacterial test of a compound of chitosan–silver particles (CS–Ag) that uses as its experimental model a native bacterium from seawater.

## 2. Materials and Methods

The following chemicals were used: chitosan (92.16% deacetylation degree) was produced from shrimp waste via alkaline hydrolysis, and with different molecular weights: high (501.59 kDa), medium (322.17 kDa) and low (130.39 kDa), as reported in a previous study [32]. Silver nitrate (AgNO_3_, >99.9%) and sodium tripolyphosphate (TPP, >86%) were purchased from Fermont (Monterrey, Mexico). Polysulfone Udel P-3500 MB7 (in pellet form, Solvay Advanced Polymers), 1-methyl-2 pyrrolidinone (NMP, >99.5%, Sigma-Aldrich, Darmstadt, Germany) and N, Ndimethylformamide (DMF, >99%, Sigma-Aldrich) were used to make polysulfone (PS) sheets used as membrane support by the phase inversion method. Trimesoyl chloride (TMC, C_9_H_3_Cl_3_O_3_), m-phenylenediamine (MPD, C_6_H_8_N_2_), sodium hydroxide (NaOH, >97%) and n-hexane (C_6_H_14_, >95%) from Sigma-Aldrich, Darmstadt, Germany, were used for the interfacial polymerization of polyamide (PA) on PS support.

### 2.1. CS–Ag Production

The production of CS–Ag was performed using the method of Ali et al. [33], with modifications (Figure 1). A solution of sodium tripolyphosphate (TPP) of 0.1% *w*/*v* (100 mL) was dropped into a chitosan solution of 0.01% *w*/*v* (250 mL) under constant agitation at 1200 rpm for 20 min, and under a flow of 10 mL min^−1^. Once the TPP solution finished dropping, a 1-millimolar silver nitrate solution (10 mL) was dropped into the suspension, which was continuously stirred for 20 min. The particles were centrifuged (Corning Life Sciences Corning LSE, Corning, NY, USA) at 6000 rpm for 40 min, and two washes with distilled water were carried out; they were then centrifuged for another 20 min. The production of CS–Ag was completed; for further characterization, samples were lyophilized at −43 °C via freeze-drying in a LabConco FreeZone (Kansas, MO, USA). This method was reproduced for high, medium and low chitosan molecular weight samples. The identification of the samples was as follows: CS–Ag-H, CS–Ag-M and CS–Ag-L, respectively.

### 2.2. Synthesis of Thin Film Composite Membranes (MCS–Ag) and Chitosan–Silver (CS–Ag) Incorporation

The thin film composite membranes were modified by incorporating CS–Ag following the method of Garcia et al. [30,34] (Figure 2). The polysulfone (PS) layer support was synthesized by the phase inversion method [34,35]. A casting solution was prepared by dissolving polysulfone pellets (16% wt) in a 4:1 DMF/NMP mixture that was stirred for 2 h at 65 °C. Afterwards, the casting solution was spread uniformly on a glass plate using a film-casting knife (BYK, Geretsried, Germany) with a clearance set at 200 μm. The PS support was immersed into a deionized (DI) water bath for 1 min at room temperature, and then washed with distilled water for 24 h. The PA layer was synthetized by the interfacial polymerization method on the PS support [13,35]. The PS support was immersed in an aqueous MPD (2% wt) solution for 2 min containing 0.05% wt NaOH. Hereafter, the membrane was immersed in TMC (0.2% wt) hexane solution for 1 min. Subsequently, the membrane was cured at 78 °C for 8 min. Finally, the PA membrane obtained was washed with distilled water and dried at room temperature for 24 h. The modified membranes were synthesized by adding each of CS–Ag-H (0.5% wt), CS–Ag-M (0.5% wt) and CS–Ag-L (0.5% wt), then homogeneously dispersed in a MPD solution with an ultrasonic bath for 1 h to prepare CS–Ag entrapping of PA layers on the porous PS supports. The modified membranes are referred to as MCS–Ag-H, MCS–Ag-M and MCS–Ag-L, respectively, while the unmodified membrane is referred to as uncoated.

### 2.3. Characterization of Chitosan–Silver Particles (CS–Ag) and Uncoated and Coated Membranes (MCS–Ag)

The analysis of Fourier transform infrared spectroscopy (FTIR) was performed for chitosan–silver particles (CS–Ag), in order to identify the functional group using a Fourier transform infrared spectrometer (Thermo Scientific Spectrum model Nicolet iS5, Waltham, MA, USA). The sample was mixed with KBr at a ratio of 0.02:0.45 (sample:KBr). The powdered sample and the KBr were previously dried at 110 °C. The mode of transmission that was applied to make the measurements was the accessory iD1 transmission. The region ranged was from 600 to 4000 cm^−1^. For membranes (uncoated and MCS–Ag), the FTIR spectra were obtained under attenuated total reflectance (ATR) mode. An average of 16 scans were attained to yield the spectrum of each sample.

The morphology of the surface of the CS–Ag was examined with a high-resolution scanning electron microscope (HR-SEM, JEOL JSM-7600F, Tokyo, Japan) that was connected to an energy-dispersive X-ray spectroscope (EDX, Oxford INCA (Abingdon, UK)), using low-angle backscatter electron imaging (LABE) to identify the elemental analysis. The acceleration voltage used was 15 kV to acquire micrographs at 5000 and 10,000×. The samples were coated with a layer of gold and palladium for 20 s, using a sputter coater (Q150R ES Quorum, Sussex, UK)). Thermogravimetric analyses (TGA) for CS–Ag were determined by using a Discovery TA Instrument, New Castle, DE, USA. The conditions of the analyses were: nitrogen flow of 25 mL min^−1^, a heating rate of 10 °C min^−1^, a temperature range of 30 to 800 °C and a sample weight of 5–10 mg. Particle size measurement and index of polydispersity (PDI) were performed by dynamic light scattering (DLS) using a Zetasizer Nano ZS (Malvern Instruments, Malvern, UK). The values of each sample were calculated as the average of five batches, using a sample concentration of 0.1 mg mL^−1^ (*v*/*v*).

The contact angle analysis for uncoated and coated membranes (MCS–Ag) was performed in Dataphysics equipment, model OCA 15EC, using the SCA20 1.0 software (dataphysics, Filderstadt, Germany). The membranes were cut into 5 cm × 2 cm pieces and placed on the base of the equipment. The measurements were made by applying five drops of water to the membrane at different locations on the surface. The surface morphology and thickness of the membranes were examined using atomic force microscopy (AFM workshop, Signal Hill, CA, USA) (model TT-AFM) in taping mode.

### 2.4. Membrane Performance Test

The desalination performance of the membranes was determined in terms of permeate flux, salt rejection and hydraulic resistance, calculated according to the method of Garcia et al. [34,36]. Prior to these tests, the membranes were soaked in distilled water for 24 h and compacted for 1 h at 2.06 MPa using distilled water. The desalination performance was evaluated in a CF042 cross-flow system (Sterlitech Corp., Auburn, WA, USA) (Figure 3) with an effective membrane area of 0.0042 m^2^ in a feed solution of 2000 and 5000 mg L^−1^ NaCl at 2.06 MPa for 300 min, based on standard test conditions [37] using a high-pressure pump of positive displacement (Hydra Cell, M03SASGSNSCA, Sterlitech Corp., Auburn, WA, USA). Pressure was obtained using pressure transmitters (PX9111, IFM, Essen, Germany). The temperature was maintained at 25 ± 1 °C using a concentric pipe heat exchanger and a benchtop chiller (PolyScience LS5, Niles, IL, USA). The permeate flux was calculated with Equation (1), as follows:(1)$$J_{v}=\frac{V}{A_{m}t}$$
where *J_v_* (L m^−2^ h^−1^) is the membrane flux, *V* (L) is the volume of permeated water, *A_m_* (m^2^) is the membrane effective area and *t* (h) is the permeation time. The membrane hydraulic resistance was calculated using Equation (2), as shown below:(2)$$R_{m}=\frac{1}{P_{m}\mu }$$
where *R_m_* is the membrane hydraulic resistance (m^−1^), *P_m_* is the membrane permeance (m Pa^−1^ s^−1^) and *µ* is water feed viscosity (Pa s). Additionally, salt rejection (*S_R_*) was measured from the feed and permeate solution. *S_R_* can be obtained using Equation (3) [38], as follows:(3)$$S_{R}=\frac{C_{f}-C_{p}}{C_{f}}\times 100$$
where *S_R_* is the salt rejection (%), and *C_f_* (g L^−1^) and *C_p_* (g L^−1^) are the concentrations of the feed solution and permeate solution, respectively. The salt permeance (*J_s_*) was calculated with the following Equation (4):(4)$$J_{s}=\frac{J_{v}}{CP}\left(\frac{100-S_{R}}{S_{R}}\right)$$
where *CP* is the concentration polarization.

### 2.5. Membrane Anti-Biofouling Test

The anti-biofouling property of the modified and unmodified membranes was tested by the immersion method in an enriched solution using as a model bacterium *Bacillus halotolerans* MCC1, which was isolated from the Sea of Cortez, Mexico, in feedwater after a typical desalination pretreatment. This solution was used to investigate its capability of bio-fouling RO membranes [12,39]. This enriched solution was elaborated following the method of Rodriguez et al. [13] and Armendariz et al. [35]. Prior to any antibacterial experiments, all supplies used were previously sterilized in an autoclave (Felisa FE-399, FELIGNEO, Jalisco, Mexico), for 15 min and 0.103 MPa at 121 °C. Bacteria were inoculated in a solution of sterilized ultrafiltered seawater (34.99 g L^−1^ salinity), peptone, glucose and yeast [38], and then incubated in a shaking incubator at 30 °C for 24 h. The prepared bacteria solution achieved a bacterial concentration of about 1 × 10^9^ CFU mL^−1^. The membrane samples were cut (2 × 2 cm), and then sterilized using ultraviolet radiation for 30 min. Then, membranes were immersed into 10 mL of bacteria solution and incubated in the shaking incubator at 30 °C for 24 h. The bacteria adhesion onto membranes was analyzed by scraping 1 cm^2^ of biofilm. Then, the biofilm cake layer thickness was measured using an inverted microscope (Zeiss Axio, Jena, Germany). A Neubauer chamber equipped with an optical microscope (Unico IP753PL, Dayton, NJ, USA) at 40× was used to carry out the total cell count of the biofilm. The amount of organic matter was determined by the ignition method [8].

### 2.6. Statistical Analyses

An analysis of variance of simple classification, based on a linear fixed effects model was carried out for each of the following variables: roughness, contact angle, membrane hydraulic resistance, permeate flux, salt rejection, salt permeance, biofilm layer thickness, number of total cells and total organic carbon. The source of variation was taken to be the type or lack of membrane coating. Once differences were detected, the means were compared using Tukey’s test for *p* ≤ 0.001. For all analyses, the professional statistical software STATISTIC 10 (StatSoft, Inc., Tulsa, OK, USA) was used.

## 3. Results and Discussion

### 3.1. Characterization of Chitosan–Silver Particles (CS–Ag) and Uncoated and Coated Membranes (MCS–Ag)

The infrared spectrum of CS–Ag-H, CS–Ag-M and CS–Ag-L material, and MCS–Ag-H, MCS–Ag-M and MCS–Ag-L, are presented in Figure 4. It is evident that all the spectra of CS–Ag (Figure 4) are remarkably similar, and the presence of the same peaks without significance differences between each other is noticeable. The peaks in the range of 2700 and 2600 cm^−1^ for the three samples are attributed to the absorption band corresponding to the C-H symmetric stretching; according to Hejazi et al. [40], a light displacement appears at around the 2800 cm^−1^ band of absorption. After the introduction of silver nitrate into chitosan, new bands at 1751.85 cm^−1^, 1751.85 cm^−1^ and 1743.72 cm^−1^ for CS–Ag-H, CS–Ag-M and CS–Ag-L, respectively, are recognizable as C=O stretching vibration [41]. The peaks for amide I (C=O vibration of amide group) are observable at 1639.44 cm^−1^ for CS–Ag-H, and the same band appeared at 1639.44 cm^−1^ for CS–Ag-M and at 1644.72 cm^−1^ for CS–Ag-L [42,43,44]. All the spectra in the range of 1400–1200 cm^−1^ are attributed to C-H bending vibration as well as C=C and N-O stretching vibration [45,46,47]. The positions found at 1059.65 cm^−1^ for CS–Ag-H and 1064.50 cm^−1^ for CS–Ag-M and 1062.20 cm^−1^ for CS–Ag-L showed the C-O-C stretching bridge of polysaccharide moieties; Rashed et al. [48] reported the same characteristic. On the other hand, the highlighted wavenumbers at 965 cm^−1^, 958.59 cm^−1^ and 963.19 cm^−1^ for CS–Ag-H, CS–Ag-M and CS–Ag-L, respectively, could refer to the bonds or –C–O–C– stretch [49]. The rises in 843.71 cm^−1^, 843.46 cm^−1^ and 845.77 cm^−1^ are assigned to C double bond CH_2_ [50]. Finally, the bands at 536.07 cm^−1^ for CS–Ag-H and 531.31 cm^−1^ for CS–Ag-M and CS–Ag-L are a characteristic of silver presence [51]. For all the membranes (Figure 4D–G), the characteristic peaks appearing at 708 cm^−1^ are attributed to the C–H bond in the aromatic group, and at 866 cm^−1^ is due to hydrogen deformation from substituting aryl groups [52]. The peaks at 1585 and 1487–1488 cm^−1^ are attributed to the stretching vibration of the aromatic ring C-C, at 1243 cm^−1^ to the asymmetric stretching C-O-C of the aryl ether group (Ar-O-Ar), and at 1169 and 1151 cm^−1^ to the symmetric stretching of the sulfone functional group (Ar-SO_2_-Ar) [53], corroborating the formation of PS membrane support. The uncoated and MCS–Ag-L present the signals at 2966–2967 cm^−1^, probably due to the amide group formation (N-H) after the interfacial polymerization reaction. This bond is not possible to observe in MCS–Ag-H and MCS–Ag-M, probably because the PA layer is not thick enough to reach the N-H wavenumber. The peaks related to the -COOH group (close to 1717 cm^−1^ and 1723 cm^−1^) are present in all membranes; however, these are weak-intensity peaks compared with CS–Ag. The combination between water and acyl chloride groups produces this functional group, which has a beneficial impact on the flux and rejection of the membranes [54].

In Figure 5, the SEM micrographs of the CS–Ag-H, CS–Ag-M and CS–Ag-L materials are visualized at 5000 and 10,000×. The CS–Ag-L (Figure 5A,B) micrographs present a base of polymer that includes small silver particles that can be observed to be overlaid on the CS matrix, and also related to that reported by the EDX analysis. This representation was also observed in the studies realized by Ashrafi et al. and Rana et al. [55,56]. The morphology of the particles is non-uniform, with a porous characteristic and the presence of cracks. The revealed fractions of silver have a defined structure, being observed as spherical, with some squares [57]. In the case of CS–Ag-M (Figure 5C,D), it is evident that there is a homogeneous distribution of particles of Ag around the surface, with a similar morphology to CS–Ag-L. The CS–Ag-H (Figure 5E,F) presents a surface with reliefs and cracks; it is possible to identify elongated particles of varied sizes that are uniformly distributed, as well as Ag particles. This is attributed to the presence of residues of TPP cross-linking agent particles in the CS during the synthesis process, with a 2.03% wt of Na according to EDX analysis. In order to corroborate the presence of Ag and to define the chemical composition of the CS–Ag, EDX analysis was conducted; the profile is illustrated in Figure 5G–I. The profile shows a signal of Ag with 15.82, 0.38 and 0.46% wt for CS–Ag-H (Figure 5I) CS–Ag-M (Figure 5H) and CS–Ag-L (Figure 5G), respectively. The elements C, N and O constitute the largest amount by weight of the particles in the three different molecular weights. In the report by Wang et al. [58], the same elements were identified. The variation between the Ag loads may be due to the molecular weight of the polymer; a higher molecular weight represents a longer polymer chain [59] which allows more active sites for placing a component such as the silver or sodium tripolyphosphate, in the moment of the cross-linking process.

The particle size distributions of CS–Ag material were characterized using dynamic light scattering (DLS), as presented in Figure 5J–L. The particle size distribution varied from 100 to 200 nm for CS–Ag-H (Figure 5L), from 120 to 350 nm for CS–Ag-M (Figure 5K) and from 80 to 180 nm for CS–Ag-L (Figure 5J). The distribution presented a wide range in size distribution due to the size of the CS structure, and in considering the three molecular weights, the minimum and maximum range of the particle size distribution was 80 and 350 nm, respectively [60,61]. In this study, the smaller particles were presented for the particles of low molecular weight; and the appreciation observed in the morphology in the SEM images (Figure 5A,B) suggested the same behavior. The indexes of polydispersity (PDI) presented in the CS–Ag were as follows: 0.431, 0.396 and 0.318 in the cases of CS–Ag-H, CS–Ag-M and CS–Ag-L, respectively. While the molecular weight decreases, the PDI value also decreases, which could be a beneficial condition for the RO membranes, since this could increase the distribution uniformity of the CS–Ag-L in the membrane surface, reducing its roughness, a condition which reduces bacterial adhesion onto modified membranes [62]. The PDI size distribution of CS–Ag was <1 but not <0.1, according to Clayton et al. [63]. When the PDI value is <0.1 it is a monodisperse sample; in this study, the CS–Ag have a broad size distribution. The same results are reported by Dara et al. [64].

The thermal stability characterization of CS–Ag was performed under a nitrogen atmosphere (Figure 6). The thermograms of CS–Ag-H, CS–Ag-M and CS–Ag-L present a weight loss (%) with an equivalent manner in the range of 30 to 125 °C. This initial loss is due to the water content [65,66]; the weight loss in this phase ranged from 8.63 to 14.25%. The lower loss weight occurred with CS–Ag-H, while in the case of CS–Ag-L and CS–Ag-M, they maintained very similar values (14.25 and 13.30%, respectively). The second phase of weight loss was reported in the range of 175 to 275 °C (25.74, 26.42 and 25.45% less for CS–Ag-H, CS–Ag-M and CS–Ag-L, respectively). This behavior is attributed to the degradation of the polymer, together with the process of depolymerization and pyrolytic decomposition of the polysaccharide [67,68]. The derivative of the analysis (DTG) is shown in Figure 6. In the range of 30 to 90 °C, the loss of water appreciates with the same behavior observed in TGA. The maximum temperature of decomposition was for CS–Ag-H (208.16 °C), CS–Ag-M (226.08 °C) and CS–Ag-L (229.12 °C); from these results, it may be inferred that while the molecular weight decreases, the maximum temperature according to the derivative of TGA increases.

The two- and three-dimensional AFM images for uncoated and MCS–Ag are shown in Figure 7, from which roughness values were obtained (Figure 8). In the images, darker areas represent the lowest points on the membrane surface, and the brighter sections show the highest points. The coated membranes show larger darker areas than the uncoated membrane. This suggests that the CS–Ag material dispersed uniformly onto the membrane surface, causing the lower peaks. This result is desirable for the desalination process because the membranes have a smaller area where bacteria present in feed water can develop and reproduce; a smooth surface makes it more difficult for bacteria to adhere to the membrane, and prevents the growth of biofilms, avoids bio-fouling and perhaps lengthens the working life of the membrane [34]. The roughness values of uncoated and coated membranes presented highly significant differences (Figure 8). The uncoated membrane shows the highest values followed by MCS–Ag-L, with statistical differences between them; the roughness values reduction are 48% (Sa) and 49% (Sq) lower than for uncoated membrane. The MCS–Ag-H and MCS–Ag-M showed the lowest values, with no statistically significant differences between them in either Sa or Sq. Those results are probably due to the cross-linking of CS, resulting in the material being well dispersed throughout the membrane surface, and covering the original valleys. A smoother membrane surface is desirable, since permeate flux under fouling conditions is maintained with decreasing roughness, and declines rapidly with increasing roughness [69].

Figure 9 shows the hydrophilicity of the membranes. It is possible to observe an increase in the contact angle in modified membranes. The lowest contact angle was obtained for the uncoated and MCS–Ag-L membranes without statistical differences between them, followed by the MCS–Ag-M and MCS–Ag-H membranes with means increased by 12.85 and 9.44%, respectively, relative to the uncoated membrane. These results may be detrimental to the desalination process, since an increase in contact angle values means a reduction in membrane hydrophilicity, thereby reducing the permeate water through the membrane; however, the contact angle result for MCS–Ag-L may be beneficial for permeate flux. These results show that the molecular weight of CS influences the membrane hydrophilicity, as higher molecular weight causes lower hydrophilicity. This probably may be due to cross-linking of the CS–Ag material, creating a dense network structure within the membrane that reduces solubility of water on the membrane surface; this generally tends to decrease water permeability through the membrane [70].

### 3.2. Desalination Performance of the Chitosan CS–Ag Modified Membrane

Figure 10 shows the results of the measured permeate flux for 2000 and 5000 mg L^−1^ NaCl. The highest flux values (43.07 ± 3.7 and 45 ± 0.82 LMH, respectively) are found for MCS–Ag-L membrane, with highly significant differences compared to the other membranes. This membrane shows increases of 56.6% and 63.3% in comparison to uncoated membrane for 2000 and 5000 mg L^−1^ NaCl, respectively. This result is likely due to the high deacetylation degree of CS used in this membrane, since lengthening the molecular chain length of the CS modifies the network structure within the membrane, thereby increasing molecular diffusion through the membrane [71]. Furthermore, this result is similar to the fluxes reported by the manufacturer of a fouling-resistant membrane for brackish water under similar working conditions [72]. The results of the other tested membranes in this study show similar values without significant differences between them. The same tendency of those membranes can be attributed to the increase in cross-link density of CS, decreasing mass transfer through the membrane [15]. Although the addition of metal nanoparticles in TFC membranes has been shown to reduce flux [35], the amount of CS–Ag added in this study was not high enough to hinder the passage of water, thus obtaining a synergy between the CS and the Ag. This result is corroborated by the contact angle results in this study (Figure 9).

Figure 11 shows the hydraulic resistance results for the membranes. An increase in resistance is observed due to the molecular weight of CS, which becomes more significant as the molecular weight is increased. The MCS–Ag-M presents the highest resistance, followed by uncoated and MCS–Ag-H, respectively, without significant statistical differences between them. These results suggest that the molecular weight of CS influences highly mass transport through the membrane, reducing (high molecular weight) or increasing (low molecular weight) the water flux (Figure 10). A higher hydraulic resistance is detrimental to the desalination process, leading to lower flux and therefore less fresh water production [73]. Moreover, the higher resistance is also correlated with lower hydrophilicity, as indicated by larger contact angle results for higher molecular weights (Figure 9).

Figure 12A shows the salt rejection for the membranes at different NaCl concentrations. A reduction in salt rejection that is associated with the NaCl concentration and the molecular weight of CS was observed. The highest salt rejection values were achieved with both the uncoated and MCS–Ag-L membranes for 5000 mg L^−1^ NaCl, without significant statistical differences between all membranes. These results suggest that material (CS–Ag-L) used in this membrane cross-linked in the structure of CS in PA while maintaining hydrophilicity of the PA membrane (Figure 9) [15]. Salt rejection was lower for 2000 than for 5000 mg L^−1^ NaCl for all membranes. Those results can be explained by the cross-linking of CS; this produces an uneven distribution of the charged species across the membrane surface, resulting in the retention of the solute particles from electrostatic repulsive forces (Coulombic attraction), in which particles approach a Gibbs–Donnan equilibrium [74]. The results in this study are in the range of dissolved salts rejection results reported by membrane manufacturers (typically 95% to greater than 99%) [75].

With regards to salt permeance through the membranes, Figure 12B shows an increase in salt passage for the coated membranes. The highest value was obtained for the MCS–Ag-L membrane, followed by MCS–Ag-H and MCS–Ag-M, without statistical differences between these last two. These results suggest that the CS–Ag material interferes with the passage of salt through the membrane. A low molecular weight of CS indicates a higher salt passage; this probably due to the CS–Ag material creating spaces in the internal structure of the membrane that are sufficiently large for salt passage, indicating that for a higher molecular weight there are more molecules that will retain the salt. However, the results indicate that the salt concentration in feed water does not influence the salt permeance for coated membranes, since for both concentrations (2000 and 5000 mg L^−1^ NaCl), the results were the same for each membrane. The uncoated membrane with 5000 mg L^−1^ NaCl showed an increase of 60% over 2000 mg L^−1^ NaCl, which suggests that the material is able to retain the salts. The salt that achieves passage may do so through the spaces created in the membrane.

### 3.3. Anti-Biofouling Test

Figure 13 presents the microphotographs of the biofilm cake layer thickness (indicated by red arrows) taken from inverted microscopy of the membranes after bactericidal and anti-adhesion experiments. The modified membranes show high bactericidal and anti-adhesion effects against *Bacillus halotolerans* MCC1 (Figure 14). It is possible to observe a similar tendency of reduction in other analyzed variables, such as biofilm cake layer (Figure 14A), total cell (Figure 14B) and total organic carbon (Figure 14C), in each membrane. Those results are due to the analyzed variables being determined from the formed biofilm on the membranes. Those results denote the high biocide and anti-adhesion capacities of membranes when CS–Ag are added in the membranes. This probably is due to the CS on the surface of the bacteria cell, which can form a polymer film that inhibits nutrient adsorption [76]. Moreover, the Ag releases Ag^+^ ions in the presence of water [77], causing bacterial death. This possible antimicrobial mechanism for *Bacillus halotolerans* MCC1 is due to this strain being Gram-positive [39]. Different biocidal mechanisms of the CS have been shown on Gram-positive and on Gram-negative strains [76]. These results prove that the CS–Ag material is toxic to *Bacillus halotolerans* MCC1, making it beneficial for desalination plants installed in regions where this strain is present. The use of CS–Ag material will reduce, control or delay bio-fouling problems. The MCS–Ag-L membrane shows the highest reduction in biofilm thickness (60%), total cell count (76%) and total organic carbon (75%), followed by MCS–Ag-M and MCS–Ag-H, compared to the uncoated membrane. These results show that reducing the CS molecular weight results in enhanced antimicrobial properties of the membranes. This is probably due to this material having smaller particle size distributions (Figure 5J–L), which increases the superficial area contact between the strain and the material, resulting in higher bacterial death. The use of low-weight CS–Ag can prevent bio-fouling and perhaps lengthen the useful life of the membrane, given that it reduces the fouling resistance of the membrane; this allows for operations to be carried out at a higher sustained flux under bio-fouling conditions.

Finally, Table 1 compares the results obtained for the MCS–Ag-L membrane in this manuscript with other NP-coated membranes reported in the literature. A significant increase can be observed in membrane permeance due to the CS–Ag coating, and the bio-fouling reduction observed is comparable to that of other materials reported in the literature, but involving a coating that is eco-friendly as well. Making this type of comparison is not simple, since the studies reported in literature were performed under different conditions, used different bacteria, coating methods and membranes.

## 4. Conclusions

In order to improve TFC-RO membranes performance, chitosan–silver particles of low, medium and high molecular weights in the range of from 80 to 350 nm were incorporated into them. The membranes were characterized by FTIR, HR-SEM, EDX, DLS and TGA. The morphology presented in SEM images was observed with chitosan as the base of polymer, and the Ag attached. The values of PDI were <1, representing a broad particle size distribution in the composite. The maximum temperature of decomposition was for CS–Ag-L (229.12 °C). The bactericidal and anti-adhesion properties of TFC-RO membranes were increased significantly through adding the CS–Ag material within the PA layer of the membranes. The molecular weight of CS–Ag significantly influences the desalination and the antimicrobial performances of the membranes; lower molecular weight leads to higher membrane performance, since it influences membrane surface properties such as hydrophilicity and roughness. The MCS–Ag-L membrane presented the highest permeate flux (up to 63%), which was attributed to the spaces in the network structure of the CS–Ag-L material; it also presented the highest antimicrobial properties (biofilm thickness reduction, total organic carbon and bacteria cell amounts). Specific features of CS–Ag-L, such as polydispersity with a particle size distribution between 80 and 180 nm, led to improved performance of the modified membrane. In order to investigate these findings further, it is advisable to carry out studies with different strains, microbial consortiums and different CS–Ag concentrations, as well as conduct testing under typical working conditions of seawater desalination plants during longer times of operation. In addition, the migration of the Ag with the use of the membranes could also shed further insights into the effectiveness of CS–Ag on RO membranes. Furthermore, it is recommended to perform tests involving the regeneration and cleaning of the modified membranes, in order to assess the susceptibility of membrane materials to chemical or physical cleaning.

The data presented here show promising results that may extend the useful life of RO membranes, since it is possible to control, delay or reduce bio-fouling problems in the desalination process on a laboratory basis, whilst increasing the permeate flux and maintaining high salt rejection. The present study contributes to the enhancement of the performance of TFC RO membranes, and may lead to further practical and important applications in the membrane design process.

## Figures and Tables

**Figure 1 membranes-12-00851-f001:**
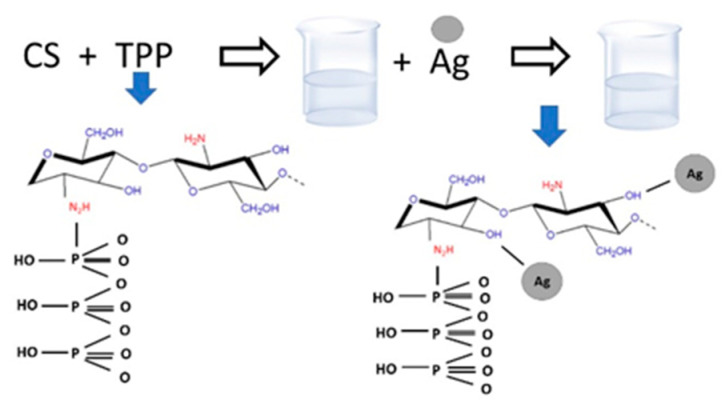
Schematic representation of the production of chitosan–silver particles.

**Figure 2 membranes-12-00851-f002:**
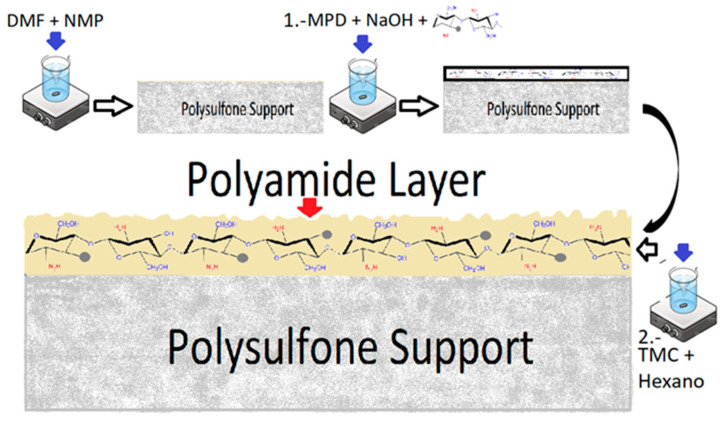
Schematic representation of the formation of membrane thin film composite (TFC) membranes by incorporating chitosan–silver particles (CS–Ag) of low, medium and high molecular weights.

**Figure 3 membranes-12-00851-f003:**
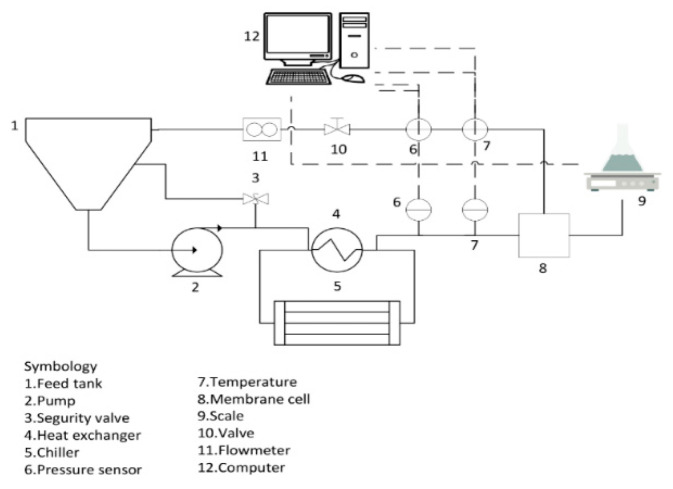
Schematic diagram of the experimental setup for testing membrane desalination performance.

**Figure 4 membranes-12-00851-f004:**
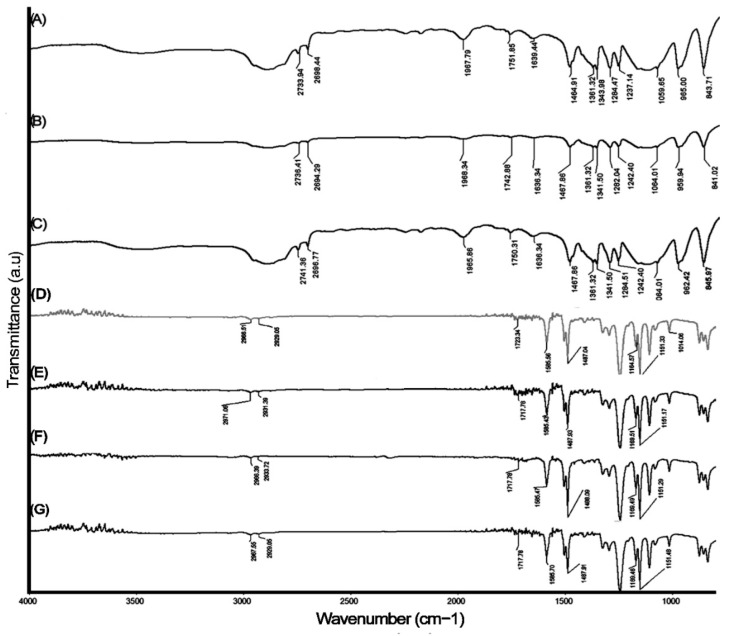
Infrared spectra of chitosan–silver particles of high ((**A**): CS–Ag-H), medium ((**B**): CS–Ag-M) and low ((**C**): CS–Ag-L) molecular weights, and infrared spectra of uncoated membranes (**D**), and those coated with chitosan–silver of high ((**E**): MCS–Ag-H), medium ((**F**): MCS–Ag-M) and low ((**G**): MCS–Ag-L) molecular weights.

**Figure 5 membranes-12-00851-f005:**
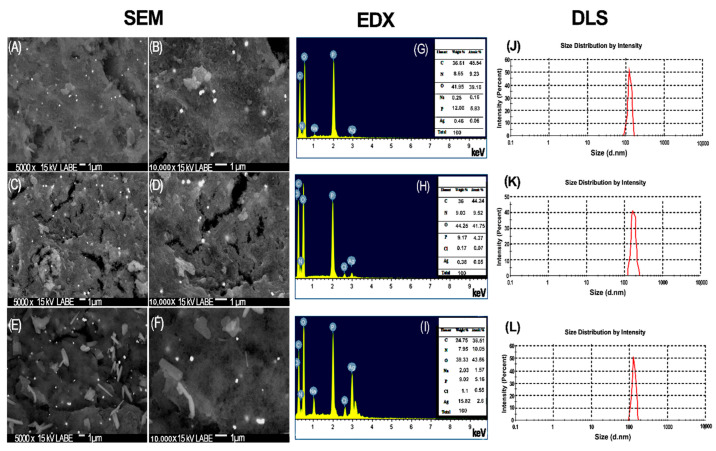
SEM of chitosan–silver particles of low (**A**,**B**), medium (**C**,**D**) and high molecular weight (**E**,**F**); EDX of chitosan–silver particles of low (**G**), medium (**H**) and high molecular weight (**I**) and DLS of chitosan–silver particles of low (**J**), medium (**K**) and high molecular weights (**L**).

**Figure 6 membranes-12-00851-f006:**
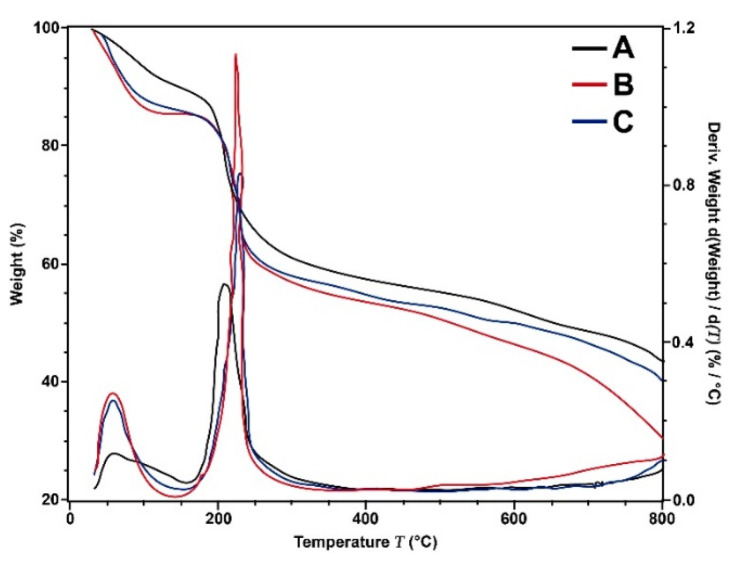
TGA and DTG thermograms of chitosan–silver particles of high ((A): CS–Ag-H), medium ((B): CS–Ag-M) and low ((C): CS–Ag-L) molecular weights.

**Figure 7 membranes-12-00851-f007:**
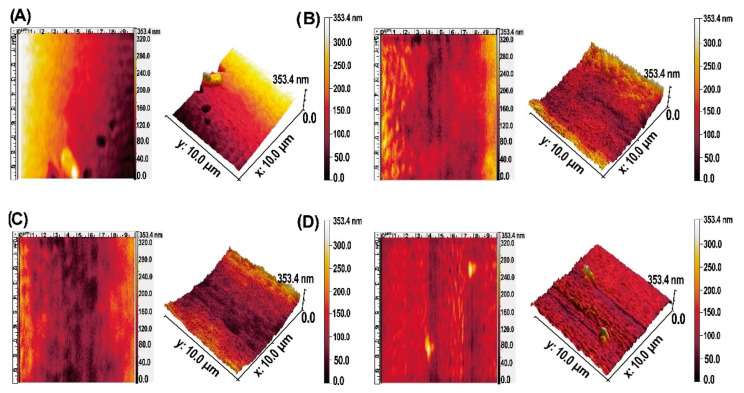
AFM images of uncoated membranes (**A**), and those coated with chitosan–silver particles of high ((**B**): MCS–Ag-H), medium ((**C**): MCS–Ag-M) and low ((**D**): MCS–Ag-L) molecular weights.

**Figure 8 membranes-12-00851-f008:**
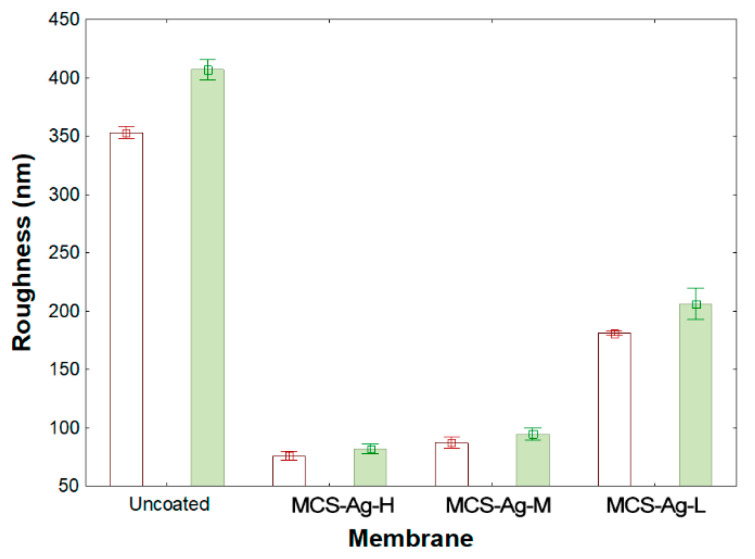
Membrane roughness results from AFM. Results are shown for uncoated membrane, as well as for membranes coated with chitosan–silver particles of high (MCS–Ag-H), medium (MCS–Ag-M) and low (MCS–Ag-L) molecular weights.

**Figure 9 membranes-12-00851-f009:**
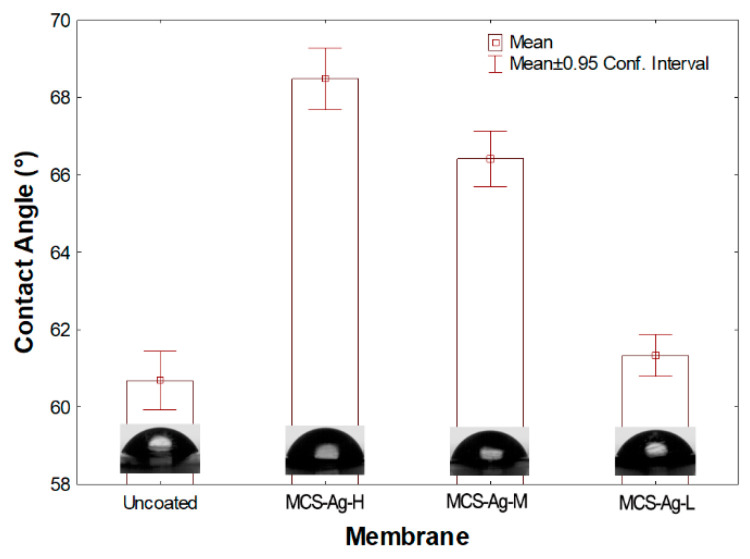
Contact angle measurements for the membranes tested. Results are shown for uncoated membrane, as well as for membranes coated with chitosan–silver particles of high (MCS–Ag-H), medium (MCS–Ag-M) and low (MCS–Ag-L) molecular weights.

**Figure 10 membranes-12-00851-f010:**
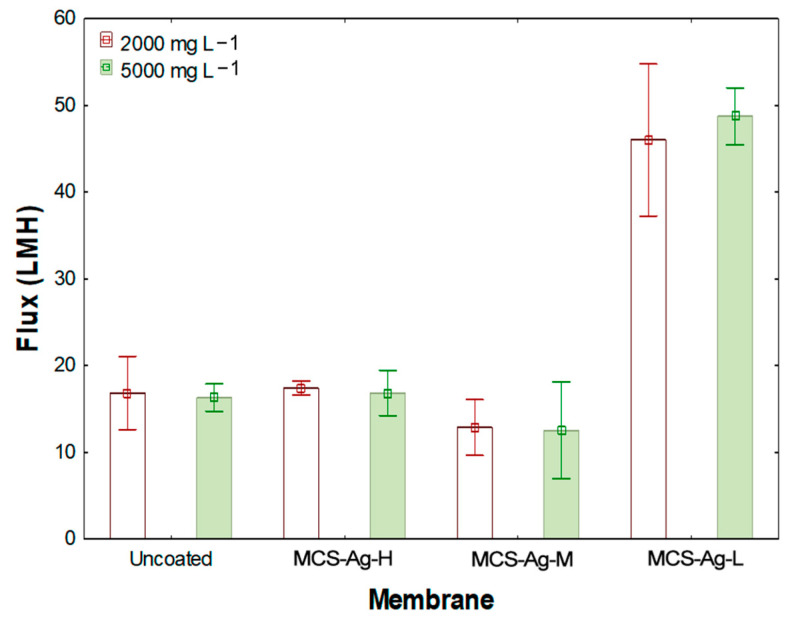
Membrane flux at different NaCl concentrations. Results are shown for uncoated membrane, as well as for membranes coated with chitosan–silver particles of high (MCS–Ag-H), medium (MCS–Ag-M) and low (MCS–Ag-L) molecular weights.

**Figure 11 membranes-12-00851-f011:**
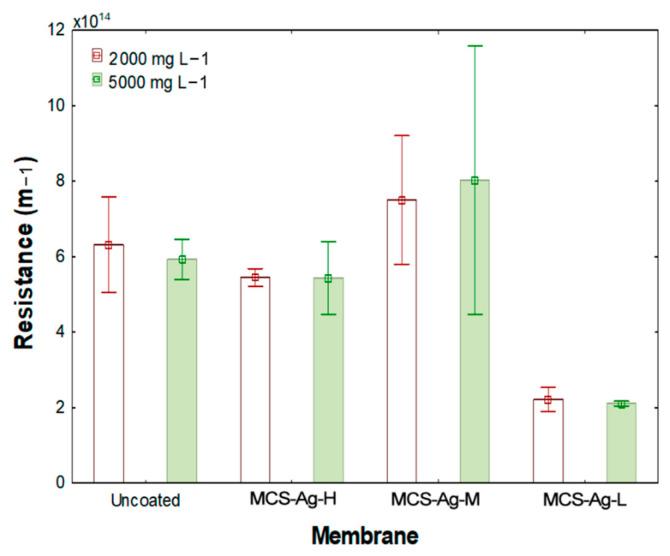
Average hydraulic resistance for membranes under standard test conditions. Results are shown for uncoated membrane, as well as for membranes coated with chitosan–silver particles of high (MCS–Ag−H), medium (MCS–Ag−M) and low (MCS–AgL) molecular weights.

**Figure 12 membranes-12-00851-f012:**
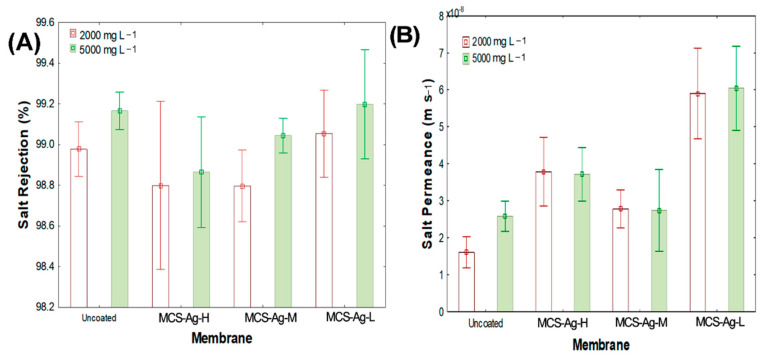
(**A**) Salt rejection and (**B**) salt permeance for the studied membranes. Results are shown for uncoated membrane, as well as for membranes coated with chitosan–silver particles of high (MCS–Ag-H), medium (MCS–Ag-M) and low (MCS–Ag-L) molecular weights.

**Figure 13 membranes-12-00851-f013:**
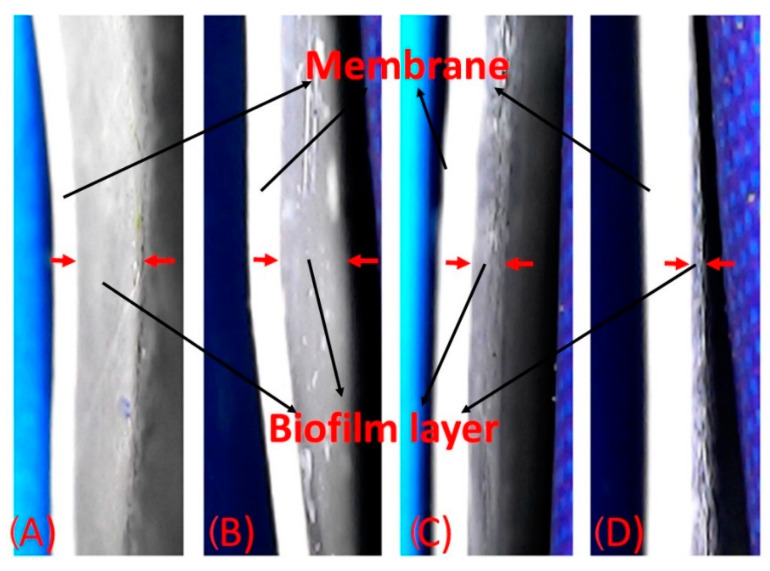
Biofilm layer thickness for membranes. Images are shown for uncoated membrane (**A**), as well as for membranes coated with chitosan–silver particles of high ((**B**): MCS–Ag-H), medium ((**C**): MCS–Ag-M) and low ((**D**): MCS–Ag-L) molecular weights.

**Figure 14 membranes-12-00851-f014:**
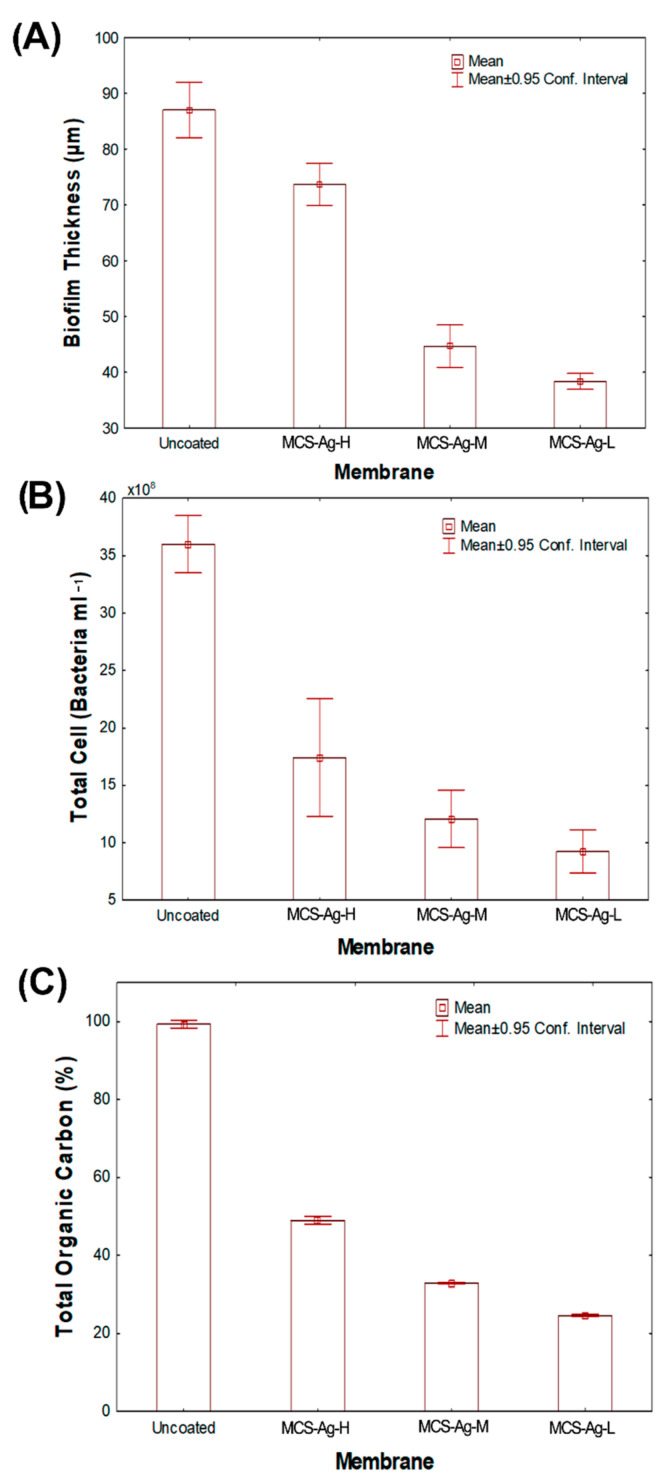
(**A**) Biofilm cake layer; (**B**) total cell count; (**C**) total organic carbon for uncoated membrane, and for membranes coated with chitosan–silver particles of high (MCS–Ag-H), medium (MCS–Ag-M) and low (MCS–Ag-L) molecular weights.

**Table 1 membranes-12-00851-t001:** Comparison of CS–Ag coated membranes with other NP-coated membranes reported in the literature.

Membrane	Coating	Coating Method	Bacteria	Desalination Working Conditions	Membrane Permeance %	Reduction of Biofilm %	Reference
TFC	CS–Ag	Interfacial polymerization	*Bacillus halotolerans* MCC1	25 °C 2.07 MPa	63	60	This study
TFC	AgNP	Interfacial polymerization	*Bacillus halotolerans* MCC1	31 °C 2.07 MPa	16.5	72	Torres-Valenzuela et al. [78]
Dow Filmtec SW30HR	FeNP	Dipping	*Bacillus halotolerans* MCC1	28 °C 6.3 MPa	41.3	44	Armendariz et al. [79]
TFC	PDA-CuNP	Electroless deposition	*E. coli*	25 °C 2.41 MPa	−39	76	Liu et al. [80]
Nitto ES20	pSM-AgNPs	ATPR	*S. paucimobilis* and *B. subtilis*	30 °C 1.5 MPa	−61.5	80 and 56	Yang et al. [81]
TFC BW30FR	PDA-CuNP	Electroless deposition	*E. coli*, *aeruginosa* and *S. aureus*	25 °C 2.41 MPa	29	68, 63 and 66	Liu et al. [82]

PDA = hydrophilic polydopamine; ATPR = atom transfer radical polymerization; pSM = poly(3-sulfopropyl methacrylate) potassium salt.

## Data Availability

Not applicable.

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
