# Peer review of "Modification of Thin Film Composite Membrane by Chitosan–Silver Particles to Improve Desalination and Anti-Biofouling Performance"

_membranes, 2022, doi:10.3390/membranes12090851_

Round 1
Reviewer 1 Report
Dear Authors
The data presented in your manuscript are very interesting for the readers and address a crucial issue in the field of desalination membranes. The research design is well organized and the explanations presented are correlated to the obtained results and characters of the developed membranes.
However, some main concerns need to be declared and additional discussion on some issues is missing.
For example;
1- The thickness of the incorporated Chitosan-Silver composite layer was not measured.
2- The effect of the modification on the mechanical properties of the modified membranes was not investigated
3- The migration of the silver nanoparticles with the use of the membranes needs to be studied and discussed.
4- The regeneration and cleaning of the modified membranes have not been studied or compared to the unmodified counterpart.
5- The effect of the developed modification on the membranes' lifetime should be studied and discussed.
In general, A major revision is needed before your manuscript can be reconsidered for publication.
Reviewer 2 Report
Comments to the Author Manuscript membranes-1839890
There are some points which must be edited or clarified by providing additional information or comments:
1. First of all, the authors need to clarify the title of the manuscript. According to the data presented in Figures 3 and 6, the size of silver particles on the membrane surface is more than 100 nm, therefore, these particles belong to the class of microparticles, but not to nano. This point should be clarified within all sections of revised manuscript.
2. The authors should write the complete terms of all abbreviations (including the instruments) before the first use in the abstract and main manuscript i.e. FTIR, HR-SEM, EDX, TGA and DLS (lines #15 and 17), in the introduction section et al.
3. It’s also recommended to add to the revised manuscript more SEM images of studied samples with higher magnification (more than 10 000x).
Did authors evaluate the loading degree of Cs-Ag particles (amount of incorporated particles). And also please discuss about uniformity of particles distribution on the membrane surface (may it will be useful to add EDX mapping images).
4. I suggest to provide more detailed study of the crystallic structure of Cs-Ag particles by XRD analysis. Presented EDX data do not provide any information about phase content of loaded silver et al…
5. Fig.7 should be improved and more readable
6. Please add to the supporting information file all images of the contact angle measurements
7. The authors are recommended to add comparison data on the similar studies of the Cs-Ag particles from the previously published papers.
8. The conclusion section should be elaborated and improved. The author should bring specific conclusions in accordance with obtained results.
Our decision on this manuscript – Major revision. After making substantial changes in article it could be recommended for publication
Reviewer 3 Report
In this manuscript, the authors designed a reverse osmosis desalination membrane by incorporation of chitosan-silver nanoparticles within the polyamide layer. The resulting membrane presented a high antibiofouling and desalination performance. This work will be helpful for materials scientists to develop new water desalination materials. I’d like to recommend the manuscript for publication once the following comments are addressed:
1. In the title of the manuscript, it should be “… improve desalination and ‘antibiofouling’ performance” instead of “…improve desalination and ‘biofouling’ performance”.
2. It might be better to draw a schematic diagram to illustrate the sample preparation process or the themes of this article, which makes it easier for readers to understand this work.
3. I recommend the authors group together the Figures of same topic for discussion, and simplify the text to highlight key points, which makes it easier for readers to understand. For example, the Figures of SEM, EDX and DLS can be grouped together to discuss the morphology and particle size of the CS-AgNPs.
4. In Figure 3, I couldn’t observe the particle shape of CS-AgNPs, it’s less like a particle and more like a polymer, however, in Figure 6, you claimed that the particle size distribution of CS-AgNPs varied from 80 to 350 nm, which is confusing. Could you explain why this is? Besides, why did CS-AgNP-H possess higher Ag particles loading rate? It was ~30 times that of the other samples, more discussion is required here.
5. In Figure 9, it might be better to add the images of the sample contact angle measurements, so that the readers can observe their hydrophilicity more intuitively.
6. Could you provide a quantitative comparison of the antibiofouling and desalination performance between your MCS-AgNPs with recently reported RO desalination membranes?
7. On page 10, line 361, it should be ‘Figure 8’ instead of ‘Figure 9’; Besides, on line 337, it should be “…and 80 to 180 nm for CS-AgNP-L”. Please read the full text carefully and correct similar errors.
8. The authors could add the following references which would again increase the interest to general distillation membrane readers: Science Advances, 2020, 6, eabb4696; Polymer, 2021, 217, 123464.
Reviewer 4 Report
Researchers systematically characterized the novel RO membranes prepared with Chitosan and Ag nanoparticles. The manuscript is well organized, and most characterizations are well explained. However, it is not ready to be published until the following concerns are properly addressed.
1. In the introduction, the author claimed that “CS can help reduce fouling because it can adsorb organic matter without losing permeance”. Will the pores of the CS membrane be clogged because of the adsorption and how can the permeance maintain the same?
2. Confusing sentence: lines 53-55 “They enhanced membrane anti-bacterial activity Escherichia coli and Staphylococcus aureus. Also, they improved membrane hydrophilicity, water flux, permeance, salt rejection, and chlorine resistance”. Please rephrase the cited research in a proper way.
3. There is an extra space in line 115.
4. In the morphology characterization, why did only CS-AgNP-H present the large quantity of TPP residues particles in the CS NPs? Furthermore, why the Ag composition in CS-AgNP-H, CS-AgNP-M, and CS-AgNP-L is different while the preparation process was the same?
5. Symbols in Figure 5 are hard to read. It would be great if the author can use color or text labels to differentiate the samples.
6. Again, a confusing sentence in lines 336-337: “The particle size distribution varied from 336 100 to 200 nm for CS-AgNP-H and 120 to 350 nm for CS-AgNP-M, and 80 to 180 nm”. Please correct.
7. What is the mechanism that “the smaller particles were presented for the nanoparticles of low molecular weight”? Also, it is not sufficient to say “morphology of SEM (Figure 3 A-B) suggested the same behavior” as the SEM images did not show obvious size difference of nanoparticles with H, M, and L molecular weight.
8. Could the author further explain why “While the molecular weight decreases the PDI value also decreases, which could be a beneficial condition for the stability in the production of RO membranes”?
9. There is a lack of pore size and pore size distribution related characterization of the prepared membranes.
10. Typo in Figure 13 text. “Salt permeance uncoated membranes” is not Grammarly correct.
11. Figure 14 looks confusing. It will be great to have any text label to indicate the membrane layer and biofilm layer.
12. Author may want to reconsider the conclusion in lines 478-480 that “These results show that reducing the CS molecular weight resulted in enhanced antimicrobial properties of the membranes”. Enhanced biofouling performance can result from changing a series of membrane properties like greater hydrophobicity, higher membrane zeta potential, smoother membrane surface, more uniform pore size distribution, etc. Reducing CS molecular weight is a method while it is not directly and necessarily linked to the anti-biofouling property. Please refer to the following references for details and cite them if needed.
· Mengying Yang, Sarah Lotfikatouli, Yvonne Chen, Tony Li, Hongyang Ma, Xinwei Mao, Benjamin S. Hsiao, Nanostructured all-cellulose membranes for efficient ultrafiltration of wastewater, Journal of Membrane Science, 650, 2022, 120422. https://doi.org/10.1016/j.memsci.2022.120422
· Mengying Yang, Pejman Hadi, Xuechen Yin, Jason Yu, Xiangyu Huang, Hongyang Ma, Harold Walker, Benjamin S. Hsiao, Antifouling nanocellulose membranes: How subtle adjustment of surface charge lead to self-cleaning property, Journal of Membrane Science, Volume 618, 2021, 118739. https://doi.org/10.1016/j.memsci.2020.118739.
Round 2
Reviewer 1 Report
Dear authors
Thank you very much for your replies.
I can recommend your revised version in its current form for publication.
Reviewer 2 Report
I still not satisfied with some of the authors' responses to the my comments. In particular, to my quotes to provide additional SEM images with more higher quality and resolution as well as EDX mapping. My strong requirement for the addition of higher resolution SEM images is due to the need to confirm the nanoscale scale of the silver particles. I remain of the opinion that in this study we have not nano, but microparticles. And the authors did not provide any evidence to refute my remark. I think that the answers of the authors are not convincing and sufficient to accept for publication this article with a deliberately low level of scientific material. If the authors do not have the opportunity to finalize the article, taking into account all the requirements of the reviewers, so that it meets the high level of the Membrane journal, the article must be withdrawn or rejectedAuthor Response
Please see the attachment

Round 3
Reviewer 2 Report
Now the manuscript is looks more logical. I just suggest to increase fig5 up to the readable scale.